# PiCa: Parameter-Efficient Fine-Tuning with Column Space Projection

**Junseo Hwang**[*]   **Wonguk Cho**[*]   **Taesup Kim**[†]
Graduate School of Data Science, Seoul National University

## Abstract

Fine-tuning large foundation models is essential for building expert models tailored to specialized tasks and domains, but fully updating billions of parameters is computationally prohibitive. Reducing the number of trainable parameters using Parameter-Efficient Fine-Tuning (PEFT), such as Low-Rank Adaptation (LoRA), is therefore crucial not only to reduce training costs but also to mitigate storage, caching, and serving overheads during deployment. Prior works, such as Singular Vectors-guided Fine-Tuning (SVFT), have shown that exploiting the geometry of pre-trained weights based on Singular Value Decomposition (SVD) can significantly improve parameter-efficiency, but they lack a solid theoretical foundation. In this paper, we introduce Parameter-Efficient Fine-Tuning with Column Space Projection (PiCa), a novel theoretically grounded PEFT method. We prove that projecting gradients onto the principal column space of pre-trained weights provides an effective inductive bias for adaptation and further enhance parameter efficiency through a novel weight-sharing strategy. Across diverse NLP and vision tasks, PiCa consistently outperforms state-of-the-art baselines under comparable or smaller parameter budgets, demonstrating both theoretical rigor and practical effectiveness.

## 1 Introduction

Fine-tuning large foundation models is essential for building expert models tailored to specialized tasks and domains. However, fully fine-tuning billions of parameters is often computationally prohibitive in terms of both training and deployment cost. Parameter-Efficient Fine-Tuning (PEFT) (Houlsby et al., 2019) addresses this challenge by adapting models with only a small number of trainable parameters while keeping the pre-trained backbone frozen. In particular, minimizing the number of trainable parameters is critical in practical scenarios where multiple adapters must be deployed simultaneously (Chen et al., 2024). In such cases, numerous sets of fine-tuned parameters for different tasks, models, and checkpoints per user must be stored separately from the pre-trained models, leading to significant storage, caching, and serving overheads.

A prominent line of research is Low-Rank Adaptation (LoRA) (Hu et al., 2022), known for its simplicity and strong empirical performance. While reducing its rank lowers the number of trainable parameters, it inevitably causes significant performance degradation. To address this, DoRA (Liu et al., 2024a) introduces weight decomposition into LoRA, achieving stronger performance at a fixed rank and often matching or surpassing LoRA while requiring only half the trainable parameters. VeRA (Kopiczko et al., 2023) further reduces parameter budgets by training small scaling vectors, demonstrating that comparable or superior performance to LoRA can be obtained with up to $4\times$ fewer trainable parameters. However, these LoRA-based methods typically rely on randomly initialized low-rank matrices and thus do not explicitly leverage the geometry or prior knowledge encoded in the pre-trained weights.

Furthermore, recent studies (Lingam et al., 2024; Han et al., 2023; Mantri et al., 2025) have shown that leveraging the geometry of pre-trained weights, particularly their spectral structure, can lead to further parameter-efficiency without performance degradation. For instance, Singular Vectors-guided Fine-Tuning (SVFT) (Lingam et al., 2024) constructs a sparse, weighted combination of a model's

---

[*]Equal contribution.
[†]Corresponding author.

pre-trained singular vectors to achieve strong performance with fewer trainable parameters. However, despite their empirical success, these SVD-based approaches (Lingam et al., 2024; Han et al., 2023; Mantri et al., 2025) lack theoretical foundation for their approaches and leave open why using the spectral structure of pre-trained weights constitutes an effective inductive bias for fine-tuning.

In this work, we propose **P**arameter-**E**fficient Fine-Tuning with **C**olumn Sp**a**ce Projection (**PiCa**), *a new theoretically grounded PEFT method that leverages the geometry of pre-trained weights.* Our theoretical analysis demonstrates that projecting gradients onto the principal column space spanned by pre-trained weights can lead to effective adaptation. This gradient projection is effectively paired with our novel weight-sharing method for further parameter efficiency. With this approach, we can significantly reduce the number of trainable parameters, even using less than the most parameter-efficient configurations of other methods (e.g., rank-1 LoRA and DoRA), while achieving significantly better performance. Our extensive experiments across various models and datasets demonstrate that PiCa consistently outperforms all baseline methods under comparable parameter budgets, as shown in Fig 1.

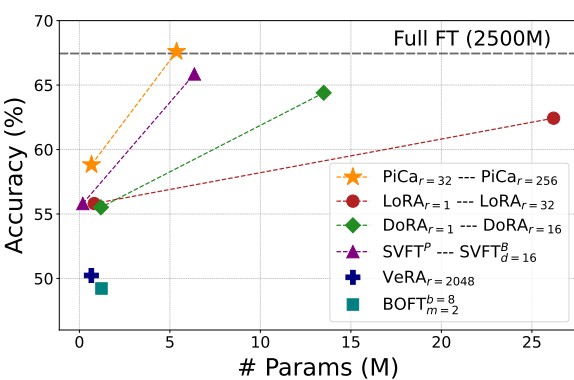

Figure 1: Average accuracy as a function of the number of trainable parameters on Commonsense Reasoning datasets using Gemma-2B. PiCa demonstrates superior performance compared to baseline methods with similar parameter budgets.

Our contributions can be summarized as follows:

- We introduce **PiCa**, a **theoretically grounded PEFT method** that explicitly exploits the geometry of pre-trained weights. We provide **a theoretical foundation** showing that projecting gradients onto the principal column space of pre-trained weights enables effective adaptation. For further parameter efficiency, PiCa also introduces a **novel weight-sharing approach** that can be paired with gradient projection.

- **PiCa consistently achieves competitive or superior performance with significantly fewer parameters** compared to other baselines. In particular, it **outperforms state-of-the-art baselines**, $SVFT^R$ and $SVFT^B$, across all datasets and models under smaller parameter budgets.

- Our experiments span a **wide range of NLP tasks** including mathematical reasoning, commonsense reasoning, and natural language understanding with different language models, as well as **diverse vision tasks** such as visual adaptation on 19 VTAB datasets with vision transformers and subject-driven generation on DreamBooth with text-to-image diffusion models. We also conduct **comprehensive ablation studies** to better understand the individual components of our method and their effects.

## 2 RELATED WORK

**Parameter-Efficient Fine-Tuning** In adapting large foundation models for downstream tasks, while full fine-tuning often yields superior performance on these tasks, its prohibitive computational overheads have motivated the development of various PEFT methods that aim to achieve comparable performance with much smaller number of trainable parameters. Recently highlighted approaches include low rank approximation (Hu et al., 2022; Liu et al., 2024a; Kopiczko et al., 2023), orthogonal reparametrization (Qiu et al., 2023; Liu et al., 2024b), and Singular Value Decomposition (SVD)-based approaches (Lingam et al., 2024; Han et al., 2023; Mantri et al., 2025).

In particular, LoRA and its variants (Hu et al., 2022; Liu et al., 2024a; Kopiczko et al., 2023) have significant attention due to its simplicity and efficiency, based low-rank decomposition. DoRA (Liu et al., 2024a) decomposes weights and achieves stronger performance at a fixed rank, often matching or surpassing LoRA while requiring only half the trainable parameters. VeRA (Kopiczko et al., 2023) further reduces parameter budgets by training small scaling vectors.

On the other hand, methods leveraging the structure of pre-trained weights, specifically through their SVD components, have been explored (Lingam et al., 2024; Han et al., 2023; Mantri et al., 2025; Meng et al., 2024). SVFT (Lingam et al., 2024) utilizes the entire singular vectors of pre-trained weights as a basis and employs a sparse matrix for updates. SVDiff (Han et al., 2023) has demonstrated fine-tuning only the singular values of pre-trained weight matrices is effective in personalization of text-to-image diffusion models. Similarly, DiTASK (Mantri et al., 2025) has shown that preserving singular vectors and enabling task-specific adaptations through neural diffeomorphic transformations of the singular values can be effective for dense prediction tasks.

Although these SVD-based methods have shown empirical success, they often lack a strong theoretical foundation that provides an analytical justification for their methods, and only few works has attempted to analyze the change in spectral structure after fine-tuning (Shuttleworth et al., 2024). In contrast, we develop a method based on a theoretical proof that the optimal rank-$r$ approximation of $\Delta W$ can be achieved by the singular vectors of the pre-trained weights, which aligns with our empirical findings. The effectiveness of this approach is validated through extensive experiments.

**Weight-sharing** Prior research has explored weight-sharing to reduce the number of parameters in neural networks (Press & Wolf, 2017; Inan et al., 2016). More recently, this concept of weight-sharing has been adapted within the LoRA framework (Kopiczko et al., 2023; Renduchintala et al., 2023; Zhou et al., 2025; Shen et al., 2024; Song et al., 2024). For instance, VeRA (Kopiczko et al., 2023) introduces a frozen random projection matrix shared across all layers, combined with trainable scaling vectors. Furthermore, recent works (Renduchintala et al., 2023; Song et al., 2024) explore different strategies of combining freezing, training, and sharing both projection matrices and scaling vectors. While demonstrating progress in parameter reduction, these prior approaches tend to be highly sensitive to randomly initialized projection matrices and often their performance is below that of standard LoRA. However, in PiCa, we construct projection matrix based on structure of pre-trained weights for each layer and share trainable weights across layers with the same function role. This approach allows significant reduction of trainable parameters without performance degradation.

## 3 METHODOLOGY

In this section, we introduce our novel PEFT method, PiCa. (1) We first discuss how fine-tuning relates to singular vectors and introduces Theorem 1, which shows that the principal subspace of pre-trained weights offers an effective space for adaptation (Section 3.1). (2) We develop this idea in the context of PEFT settings, showing that sequentially projecting gradients onto this subspace offers a theoretically grounded way to perform fine-tuning under parameter constraints through Theorem 2 (Section 3.2). (3) On top of these insights, we finally present our algorithm, PiCa, which integrates sequential projection with weight-sharing for further parameter-efficient adaptation (Section 3.3).

### 3.1 FINE-TUNING AND COLUMN SPACE PROJECTION

"Fine"-tuning is, by definition, the process of making a relatively small update from the pre-trained weights $W_0$ to the fine-tuned weights $W^*$, in order to adapt a model to a specific downstream task with a much smaller dataset. As large foundation models are pre-trained on vast, diverse corpora, good optima tend to lie in a small-update neighborhood of $W_0$. Therefore, in the context of fine-tuning of large foundation models, it is natural to assume that $\Delta W = W^* - W_0$ with $\|W_0\| \gg \|\Delta W\|$. Lemma 3.1 indicates that, when this change is small, the leading singular structures of $W_0$ and $W^*$ remain closely aligned.

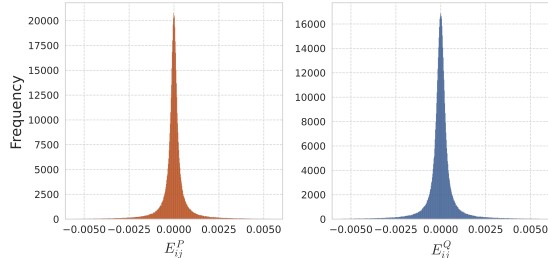

Figure 2: Distribution of $E_{ij}^P$ and $E_{ij}^Q$ across all weight matrix elements using DeBERTaV3$_{\text{base}}$. Most values are tightly concentrated around zero, validating that $\mathcal{O}(\epsilon)$ is negligible in practice.

**Lemma 3.1** (Wedin (1972))**.** *Let $W_0, W^* \in \mathbb{R}^{m \times n}$ with $W^* = W_0 + \Delta W$. Let $U_r, U_r^*$ denote the top-$r$ left singular-vector matrices of $W_0$ and $W^*$. Define the gap*

$$\delta := \min \Big\{ \sigma_r(W_0) - \sigma_{r+1}(W^*), \ \sigma_r(W^*) - \sigma_{r+1}(W_0) \Big\}.$$

*Then for any unitarily invariant norm,*

$$\| \sin \Theta(U_r, U_r^*) \| \ \leq \ \frac{\|\Delta W\|}{\delta}.$$

Building on this insight, Theorem 1 expresses the relation between $W_0$ and $W^*$ in a form that involves a small deviation $E$, and uses this to analyze how the update $\Delta W$ can be captured within the column space of $U_r$. Empirical results in Fig. 2 support this view, showing that the entries of $E$ are tightly concentrated near zero.

**Theorem 1** (Approximation error of projection onto $U_r$)**.** *Let $W_0 = U \Sigma V^\top \in \mathbb{R}^{m \times n}$ be the Singular Value Decomposition (SVD) of $W_0$. Suppose the fine-tuned matrix $W^* \in \mathbb{R}^{m \times n}$ has the form*

$$W^* = (UP)\Sigma^*(VQ)^\top,$$

*where:*

- *$U^* = UP$ and $V^* = VQ$ are the left and right singular vectors of $W^*$, respectively,*

- *$\Sigma^* = diag(\sigma_1(W^*), \ldots, \sigma_{\min(m,n)}(W^*))$,*

- *$P = I_m + E^P$, $Q = I_n + E^Q$, with $|E_{ij}^P| < \epsilon$, $|E_{ij}^Q| < \epsilon$.*

*Let $\Delta W = W^* - W_0$, and let $U_r \in \mathbb{R}^{m \times r}$ be the top-$r$ left singular vectors of $W_0$. Then, the approximation error incurred by projecting $\Delta W$ onto the subspace spanned by $U_r$ satisfies*

$$\left\| \Delta W - U_r U_r^\top \Delta W \right\|_F^2 \leq \sum_{i=r+1}^{\min(m,n)} \sigma_i^2(\Delta W) + \mathcal{O}(\epsilon).$$

The complete proof of Theorem 1 is detailed in Appendix B.

Theorem 1 indicates that the update $\Delta W$ can be well captured within the principal column space of $W_0$. The first term on the right-hand side, $\sum_{i=r+1}^{\min(m,n)} \sigma_i^2(\Delta W)$, corresponds to the rank-$r$ approximation error of $\Delta W$ given by the Eckart–Young theorem (Eckart & Young, 1936). The additional $\mathcal{O}(\epsilon)$ term reflects the small deviation introduced through $E^P$ and $E^Q$, and empirical evidence in Fig. 2 suggests that the $\mathcal{O}(\epsilon)$ term is negligible in practice. Appendix C.2 provides further observations on large-scale models, which is consistent with this view.

Theorem 1 shows that the dominant directions of the resulting update $\Delta W$ are well captured within the pre-trained column space $U_r$ of $W_0$. This implies that by keeping $U_r$ fixed and learning only a small set of coefficients that determine the task-specific choice of how to move inside this space, we can substantially reduce the number of trainable parameters, which is precisely the notion of parameter efficiency we target.

Theorem 1 is not meant to show that $U_r$ projection is globally optimal or that projection alone guarantees task-optimal performance. Other projection spaces may also reach good optima, which does not contradict our claims. Rather, our contribution is to provide theoretical support for why this particular projection can work well, whereas most prior methods are justified only empirically.

### 3.2 SEQUENTIAL GRADIENT PROJECTION

Theorem 2 shows that the principal column space in Theorem 1 can be naturally incorporated into PEFT by projecting gradients onto the subspace at each step. This provides a practical way to exploit the same effective space throughout training, offering a simple and theoretically supported view of how sequential updates can operate within the projection framework.

---

**Algorithm 1:** Adam with PiCa

---

**Input:** rank $r$; learning rate $\eta$; decay rates $\beta_1, \beta_2$; small $\varepsilon > 0$.

**Setup / Notation.** For each group $f \in \mathcal{F}$ and layer $i$: compute SVD $W_0^{f,i} = U^{f,i}\Sigma^{f,i}(V^{f,i})^\top$
  and set $P^{f,i} \leftarrow U_{[:,1:r]}^{f,i}$ ;                   // Layer-wise fixed projector
Set $W^{f,i} \leftarrow W_0^{f,i} \in \mathbb{R}^{m \times n}$ ;
For each group $f$: set shared compact states $B_0^f, M_0^f, V_0^f \in \mathbb{R}^{r \times n} \leftarrow 0$; set $t \leftarrow 0$;
*Elementwise ops:* $\odot$ (Hadamard), $\oslash$ (elementwise divide), $\sqrt{\cdot}$ (elementwise).

**repeat**
$\quad$ $t \leftarrow t + 1$ ;
$\quad$ **foreach** *group* $f$ **do**
$\quad\quad$ // (1) Project layer-wise gradients & aggregate
$\quad\quad$ $R_t^f \leftarrow \sum_i \; (P^{f,i})^\top \left( -\nabla_{W^{f,i}} \ell_t(W^{f,i}) \right)$ ;
$\quad\quad$ // (2) Adam update in compact space
$\quad\quad$ $M_t^f \leftarrow \beta_1 M_{t-1}^f + (1 - \beta_1) R_t^f$ ;
$\quad\quad$ $V_t^f \leftarrow \beta_2 V_{t-1}^f + (1 - \beta_2)(R_t^f \odot R_t^f)$ ;
$\quad\quad$ $\hat{M}_t^f \leftarrow M_t^f/(1 - \beta_1^t); \quad \hat{V}_t^f \leftarrow V_t^f/(1 - \beta_2^t)$ ;
$\quad\quad$ $\Delta B_t^f \leftarrow \hat{M}_t^f \oslash (\sqrt{\hat{V}_t^f} + \varepsilon); \quad B_t^f \leftarrow B_{t-1}^f + \eta \, \Delta B_t^f$ ;
$\quad\quad$ // (3) Decompress shared update to each layer
$\quad\quad$ **foreach** *layer* $i$ **do**
$\quad\quad\quad$ $W^{f,i} \leftarrow W^{f,i} + \eta \; P^{f,i} \; \Delta B_t^f$ ;

**until** *convergence*;
**return** $\{B_T^f\}_{f \in \mathcal{F}}$ ;                   // Final shared compact parameters

---

**Definition 1** (L-smoothness for matrix-valued functions)**.** A differentiable function $\ell : \mathbb{R}^{m \times n} \to \mathbb{R}$ is *L-smooth (w.r.t. $\|\cdot\|_F$)* if

$$\|\nabla \ell(W_1) - \nabla \ell(W_2)\|_F \; \leq \; L \, \|W_1 - W_2\|_F \quad \text{for all } W_1, W_2 \in \mathbb{R}^{m \times n}.$$

**Theorem 2** (Sequential projection approximates accumulated projection)**.** *Let $\ell : \mathbb{R}^{m \times n} \to \mathbb{R}$ be L-smooth with $\|\nabla \ell(W)\|_F \leq G$. Define the unprojected gradient descent path*

$$Z_{t+1} = Z_t - \eta \nabla \ell(Z_t).$$

*Let the* accumulated-projection *iterate be*

$$W_T = W_0 - \eta \, \Pi_{U_r}\Big(\sum_{t=0}^{T-1} \nabla \ell(Z_t)\Big),$$

*and the* sequential-projection *iterates*

$$P_{t+1} = P_t - \eta \, \Pi_{U_r} \nabla \ell(P_t), \qquad P_0 = W_0,$$

*where $\Pi_{U_r} = U_r U_r^\top$ is the fixed rank-r projector.*

*Then, for any $T$, the difference satisfies*

$$\|W_T - P_T\|_F \; \leq \; \frac{\eta^2}{2} \, LG\,T(T-1) + O((\eta L T)^3).$$

The proof is provided in Appendix B.

### 3.3 PiCa: PEFT with Column Space Projection

Based on the preceding results, we propose PiCa that projects gradients onto the principal column space spanned by pre-trained weights for each update. This gradient projection is effectively paired with our novel weight-sharing method for further parameter efficiency. For clarity, we describe PiCa in Algorithm 1 using Adam, though the approach is not limited to this optimizer.

In Algorithm 1, each functional group $f \in \mathcal{F} = \{\text{query,key,value},\dots\}$ is associated with a single trainable matrix $B^f \in \mathbb{R}^{r \times n}$, which is shared across all layers $i = 1,\dots,L$ of the same group. The projection matrices $P^{f,i}$ remain layer-specific, leveraging the geometry of each pre-trained weight $W_0^{f,i} \in \mathbb{R}^{m \times n}$. The gradients of each layer $i$ are first projected onto $P^{f,i}$ defined by the top-$r$ singular vectors of the corresponding pre-trained weight, $U_r^{f,i} \in \mathbb{R}^{m \times r}$. The updates are then accumulated in this compact space as shared parameters $B^f$. Momentum and variance statistics are also updated in this compact space. The shared update is then mapped back to each layer through its layer-specific projector $U_r^{f,i}$.

This procedure can be implemented using the reparameterization

$$W^{f,i} = W_0^{f,i} + U_r^{f,i} B^f,$$

where $B^f$ is a trainable matrix initialized with zero and $U_r^{f,i}$ remains fixed during fine-tuning. Under this parameterization, optimizing $B^f$ is mathematically equivalent to the update in Algorithm 1.

Unlike prior approaches (Kopiczko et al., 2023; Renduchintala et al., 2023) that primarily rely on random projection matrices for weight-sharing, our method leverages layer-specific projection matrices $U_r^{f,i}$ derived from the structure of the pre-trained weights $W_0^{f,i}$ for each layer $i$ of group $f$. This allows us to capture the distinct characteristics and pre-trained knowledge encoded in each $W_0^{f,i}$. Given the use of unique projection matrices per layer, we posit that the trainable parameter $B^f$ can be effectively shared across layers with the same functionality, facilitating efficient adaptation to downstream tasks. Our extensive experiments demonstrate the effectiveness of weight-sharing in PiCa, which reduces the number of trainable parameters by up to $7\times$ without compromising performance (see Sec. 4.3 for details).

## 4 Experiments

### 4.1 Experimental settings

We evaluate the effectiveness of PiCa across a diverse set of Natural Language Processing (NLP) tasks, covering Mathematical Reasoning, Commonsense Reasoning, and Natural Language Understanding (NLU). For Mathematical Reasoning tasks, we fine-tune our model on the MetaMathQA-40K dataset (Yu et al., 2023) and assess its performance on the GSM-8K (Cobbe et al., 2021) and MATH (Hendrycks et al., 2021) datasets. Furthermore, we conduct evaluations on eight commonsense reasoning benchmarks: BoolQ (Clark et al., 2019), PIQA (Bisk et al., 2020), SIQA (Sap et al., 2019), HellaSwag (Zellers et al., 2019), Winogrande (Sakaguchi et al., 2019), ARC-Easy/ARC-Challenge (Clark et al., 2018), and OpenBookQA (Mihaylov et al., 2018). For NLU tasks, we utilize the GLUE benchmark (Wang et al., 2018). We report matched accuracy for MNLI, Matthew's correlation for CoLA, Pearson correlation for STS-B, and accuracy for all other tasks. We employ the Gemma-2B/7B (Team et al., 2024), and LLaMA-3-8B (AI, 2024) models for Mathematical Reasoning tasks and adopt the DeBERTaV3-base (He et al., 2023) model for NLU tasks.

Beyond NLP, we also evaluate PiCa on vision tasks. Specifically, we conduct experiments with visual adaptation using the ViT-B/16 (Dosovitskiy et al., 2021) on 19 different datasets of VTAB-1K (Zhai et al., 2020), grouped into *Natural*, *Specialized*, and *Structured* categories. Performance is reported as the average accuracy across these groups. In addition, we evaluate subject-driven generation tasks with the Stable Diffusion v2.1 (Rombach et al., 2022) on the DreamBooth dataset (Ruiz et al., 2023), which includes 30 subjects and 25 prompts per subject, totaling 750 different personalization tasks. Following prior work (Ruiz et al., 2023), we report results using DINO for subject fidelity and CLIP-T for text fidelity. To ensure a fair comparison, hyperparameters and training protocols are

aligned with those outlined in (Lingam et al., 2024; Cho et al., 2024). Further details are provided in the Appendix C. [1]

## 4.2 RESULTS

For a fair comparison, we follow (Lingam et al., 2024; Dosovitskiy et al., 2021; Cho et al., 2024) and evaluate the effectiveness of PiCa across three NLP tasks (Mathematical Reasoning, Commonsense Reasoning, and Natural Language Understanding) and two vision tasks (Visual Adaptation and Subject-Driven Generation). The baselines include LoRA (Hu et al., 2022), DoRA (Liu et al., 2024a), BOFT (Liu et al., 2024b), VeRA (Kopiczko et al., 2023), and SVFT (Lingam et al., 2024). Full experimental details are provided in Appendix C.

**Mathematical Reasoning**  In Table 1, we provide results on mathematical question answering, comparing our method against baseline PEFT methods across three different base models ranging from 2B to 8B parameters. Our experiments include two configurations of PiCa: a high-rank setting with fewer trainable parameters than $SVFT^R$, and a low-rank configuration with fewer trainable parameters than rank 1 LoRA. As shown in Table 1, our high-rank PiCa consistently achieves superior performance while using the fewest trainable parameters across all models and datasets. In the low-rank setting, PiCa achieves either the best or second-best performance.

Table 1: Performance on Mathematical Reasoning benchmarks (GSM-8K and MATH). #Params indicates the number of trainable parameters. The best and second-best PEFT methods are highlighted in **bold** and underlined, respectively. For Gemma-7B, we set $r = 16$ to ensure the number of trainable parameters remains below that of rank-1 LoRA. For $SVFT_d^R$, we use $d = 16$ for Gemma models and $d = 12$ for LLaMA-3 models. In the high-rank setting, PiCa consistently achieves the best performance across all models and datasets, while using the fewest trainable parameters.

| Method | Gemma-2B | | | Gemma-7B | | | LLaMA-3-8B | | |
|---|---|---|---|---|---|---|---|---|---|
| | #Params | GSM-8K | MATH | #Params | GSM-8K | MATH | #Params | GSM-8K | MATH |
| Full-FT | 2.5B | 52.69 | 17.94 | 8.5B | 78.09 | 30.98 | 8.0B | 76.57 | 26.12 |
| $BOFT_{m=2}^{b=8}$ | 1.22M | 36.01 | 12.13 | 2.90M | 71.79 | **28.98** | 4.35M | 67.09 | 21.64 |
| $DoRA_{r=1}$ | 1.19M | 35.35 | 13.04 | 3.26M | **74.37** | 26.28 | 2.55M | 68.30 | 21.96 |
| $LoRA_{r=1}$ | 0.82M | 32.97 | 13.04 | 0.82M | 72.40 | 26.28 | 1.77M | 68.84 | 20.94 |
| $VeRA_{r=1024}$ | 0.63M | 36.77 | 14.12 | 0.43M | 71.11 | 27.04 | 0.98M | 63.76 | 20.28 |
| $SVFT^P$ | 0.19M | 40.34 | 14.38 | 0.43M | 73.50 | 27.30 | 0.48M | 69.22 | 20.44 |
| $PiCa_{r=32}$ | 0.67M | **41.32** | **15.22** | 0.64M | 74.30 | 28.92 | 1.38M | **73.54** | **24.14** |
| $LoRA_{r=32}$ | 26.2M | 43.06 | 15.50 | 68.8M | 76.57 | 29.34 | 56.6M | 75.89 | 24.74 |
| $DoRA_{r=16}$ | 13.5M | 44.27 | 16.18 | 35.5M | 74.52 | 29.84 | 29.1M | 75.66 | 24.72 |
| $SVFT_d^R$ | 6.35M | 50.03 | 15.56 | 19.8M | 76.81 | 29.98 | 13.1M | 75.90 | 24.22 |
| $PiCa_{r=256}$ | 5.37M | **52.77** | **16.36** | 10.22M | **78.39** | **30.16** | 11.01M | **76.12** | **24.88** |

**Commonsense Reasoning**  In Table 2, we evaluate commonsense reasoning performance on eight benchmark datasets using Gemma-7B, following the same experimental setup as in the Mathematical Reasoning task. We compare both high-rank and low-rank configurations of our method against PEFT baselines. In both settings, PiCa outperforms all baselines on average across the eight datasets. In the high-rank setting, our method achieves state-of-the-art performance on seven out of eight datasets while using over $13\times$ fewer parameters than LoRA, and it consistently outperforms SVFT on all eight datasets with approximately half the number of parameters. In the low-rank setting, PiCa also achieves the best average performance, surpassing rank 1 DoRA while using more than $5\times$ fewer parameters. Compared to $SVFT^P$, our method delivers superior performance on seven out of eight datasets, with an average improvement of nearly two percentage points. Similar trends are observed with Gemma-2B (see Appendix C.3).

**Natural Language Understanding**  Table 3 presents the results on the GLUE benchmark using DeBERTaV3$_{base}$. Compared to LoRA with rank 8, our method achieves over one percentage point

---

[1]The official implementation is available at `https://github.com/hjunseoh/PiCa`.

Table 2: Performance on Commonsense Reasoning benchmarks. #Params refers to the number of trainable parameters. The best and second-best PEFT methods are highlighted in **bold** and underlined text, respectively. In the high-rank setting, PiCa achieves state-of-the-art performance on 7 out of 8 datasets, using over $13\times$ fewer parameters than LoRA and about half the parameters of SVFT.

| Method | #Params | BoolQ | PIQA | SIQA | HS | WG | ARC-e | ARC-c | OBQA | Avg. |
|---|---|---|---|---|---|---|---|---|---|---|
| Full-FT | 8.5B | 72.32 | 87.32 | 76.86 | 91.07 | 81.76 | 92.46 | 82.87 | 89.00 | 84.19 |
| DoRA$_{r=1}$ | 3.31M | 68.22 | **86.72** | 75.23 | 91.14 | **78.13** | 91.87 | **83.19** | **86.20** | 82.59 |
| VeRA$_{r=2048}$ | 1.49M | 64.25 | 86.28 | 74.04 | 86.96 | 69.00 | 92.76 | 82.33 | 82.00 | 79.70 |
| LoRA$_{r=1}$ | 0.82M | 65.44 | 86.28 | 75.02 | 89.91 | 75.92 | 91.79 | 81.91 | 85.40 | 81.46 |
| SVFT$_P$ | 0.51M | 67.92 | 86.45 | 75.47 | 86.92 | 74.03 | 91.80 | 81.21 | 83.00 | 80.85 |
| PiCa$_{r=16}$ | 0.64M | **70.95** | 86.29 | **76.00** | **91.42** | 76.32 | **92.89** | **83.19** | 85.60 | **82.83** |
| LoRA$_{r=32}$ | 68.8M | 71.55 | 87.95 | 77.27 | 91.80 | **79.71** | 92.67 | 82.16 | 86.40 | 83.69 |
| DoRA$_{r=16}$ | 35.5M | 71.46 | 87.59 | 76.35 | 92.11 | 78.29 | 92.00 | 80.63 | 85.60 | 83.00 |
| SVFT$^B_{d=8}$ | 9.80M | 71.90 | 86.96 | 76.28 | 91.55 | 78.76 | 92.80 | 83.11 | 85.40 | 83.35 |
| PiCa$_{r=128}$ | 5.11M | **72.84** | **87.98** | **77.79** | **92.82** | 79.40 | **93.14** | **83.62** | **88.20** | **84.47** |

Table 3: Performance of DeBERTaV3$_{base}$ on the GLUE benchmark. #Params refers to the number of trainable parameters. The best and second-best PEFT methods are highlighted in **bold** and underlined text, respectively. While using more than $2.5\times$ fewer parameters than SVFT$^R_{d=2}$, PiCa outperforms it on all datasets.

| Method | #Params | MNLI | SST-2 | MRPC | CoLA | QQP | QNLI | RTE | STS-B | Avg. |
|---|---|---|---|---|---|---|---|---|---|---|
| Full-FT | 183.83M | 89.90 | 95.63 | 89.46 | 69.19 | 92.40 | 94.03 | 83.75 | 91.60 | 88.25 |
| LoRA$_{r=8}$ | 1.33M | **90.65** | 94.95 | 89.95 | 69.82 | **93.87** | 91.99 | 85.20 | 91.60 | 88.50 |
| LoRA$_{r=1}$ | 0.17M | 90.12 | 95.64 | 86.43 | 69.13 | 91.43 | 94.18 | 87.36 | 91.52 | 88.23 |
| DoRA$_{r=4}$ | 0.75M | 89.92 | 95.41 | 89.10 | 69.37 | 91.53 | 94.14 | 87.00 | 91.80 | 88.53 |
| BOFT$^{b=8}_{m=2}$ | 0.75M | 90.25 | **96.44** | **92.40** | 72.95 | 92.10 | 94.23 | 88.81 | **91.92** | **89.89** |
| VeRA$_{r=1024}$ | 0.09M | 89.93 | 95.53 | 87.94 | 69.06 | 90.40 | 93.24 | 87.00 | 88.71 | 87.73 |
| SVFT$^P$ | 0.06M | 89.69 | 95.41 | 88.77 | 70.95 | 90.16 | **94.27** | 87.24 | 91.80 | 88.54 |
| SVFT$^R_{d=2}$ | 0.28M | 89.97 | 95.99 | 88.99 | 72.61 | 91.50 | 93.90 | 88.09 | 91.73 | 89.10 |
| PiCa$_{r=16}$ | 0.11M | 90.20 | 96.00 | 91.40 | **73.10** | 91.60 | 94.20 | **89.20** | 91.80 | 89.69 |

Table 4: Performance on vision benchmarks. VTAB-1K (ViT-B/16) is averaged over 19 datasets grouped into *Natural, Specialized, Structured*. DreamBooth is evaluated with Stable Diffusion v2.1 using DINO (subject fidelity) and CLIP-T (text fidelity). The best and second-best results are highlighted in **bold** and underlined, respectively.

| VTAB-1K (ViT-B/16) | | | | | | DreamBooth (Stable Diffusion v2.1) | | | |
|---|---|---|---|---|---|---|---|---|---|
| Method | #Params | Natural | Specialized | Structured | All | Method | #Params | DINO | CLIP-T |
| LoRA$_{r=8}$ | 1.32M | 0.823 | **0.851** | **0.508** | 0.696 | LoRA$_{r=16}$ | 3.37M | 0.618 | 0.305 |
| DoRA$_{r=8}$ | 1.41M | **0.827** | 0.846 | 0.505 | 0.695 | DoRA$_{r=16}$ | 3.42M | 0.617 | 0.306 |
| SVFT$^B_{d=8}$ | 0.93M | 0.820 | 0.844 | 0.486 | 0.684 | SVFT$^B_{d=12}$ | 2.50M | 0.622 | **0.307** |
| VeRA$_{r=4096}$ | 0.45M | 0.813 | 0.845 | 0.474 | 0.677 | VeRA$_{r=13312}$ | 1.80M | 0.613 | 0.305 |
| PiCa$_{r=64}$ | 0.44M | 0.825 | **0.851** | **0.508** | **0.697** | PiCa$_{r=128}$ | 1.72M | **0.634** | 0.306 |

higher average performance. While using more than $2.5\times$ fewer parameters than SVFT$^R_{d=2}$, our method outperforms it on all datasets. Furthermore, despite using over $7\times$ fewer parameters than BOFT, our method achieves comparable average performance.

**Vision Experiments** Table 4 reports results on VTAB-1K and DreamBooth dataset. On the VTAB-1K dataset, PiCa achieves the best overall score while using the fewest trainable parameters. In particular, PiCa achieves competitive results compared to other baselines while using 2 to $3\times$ fewer trainable parameters in VTAB-1K. On the DreamBooth dataset, PiCa achieves a higher DINO score

while maintaining a comparable CLIP-T score, demonstrating strong personalization with fewer parameters than other baselines. These results highlight that PiCa maintains strong performance on vision tasks under substantially reduced parameter budgets.

### 4.3 FURTHER ANALYSIS

**Ablation Study of Column Space Projection** In Table 5, we compare the effect of using column space projection versus random space projection. We use commonsense reasoning benchmarks with Gemma-2B. The results show that column space projection improves overall accuracy by 4.42 points compared to random space projection, demonstrating the effectiveness of leveraging the spectral structure of pre-trained weights, aligned with the results in Theorem 1.

Table 5: Ablation study on projection choice (rank = 256). Average scores are reported across commonsense reasoning benchmarks using Gemma-2B.

| Projection Method | #Params | Avg. |
|---|---|---|
| Random Space | 5.37M | 63.18 |
| Column Space (Ours) | 5.37M | **67.60** |

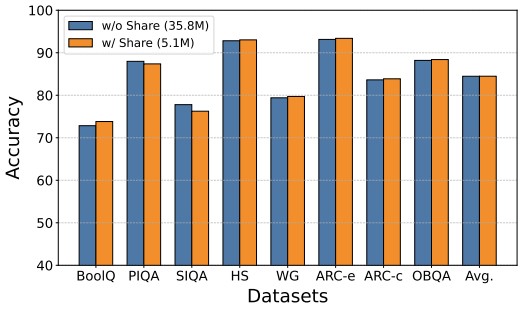

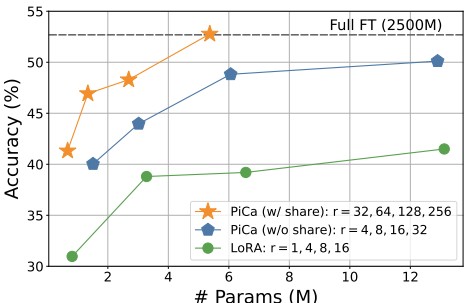

(a) Accuracy across Commonsense Reasoning datasets with and without weight-sharing. weight-sharing reduces the number of trainable parameters by up to 7× without compromising performance.

(b) Accuracy on GSM-8K under varying rank settings. weight-sharing consistently yields superior performance under similar parameter budgets.

Figure 3: Ablation study of weight-sharing across different datasets and rank configurations.

**Ablation Study of Weight-sharing** In Fig. 3a, we analyze the impact of weight-sharing in PiCa across eight Commonsense Reasoning datasets using Gemma-7B. By comparing PiCa with its standard configuration (rank 128 with weight-sharing; 5.1M trainable parameters) against a variant without sharing (rank 16; 35.8M parameters), we find that the default PiCa consistently achieves performance comparable to its non-sharing variant while requiring about 7× fewer trainable parameters.These results indicate that weight-sharing substantially improves parameter efficiency without performance degradation.

Furthermore, we conduct an additional study on the effect of weight-sharing under varying rank settings using the GSM-8K benchmark with Gemma-2B. As shown in Fig. 3b, PiCa consistently achieves superior performance under similar parameter budgets compared to both its no-sharing ablation and LoRA.

## 5 DISCUSSION

While PiCa significantly reduces the number of trainable parameters required, it introduces a minor limitation during inference. Specifically, PiCa stores only a small shared matrix $B^f$ for each functional group $f$, but requires to perform an additional SVD on the pre-trained weights $W_0$ at load time to recover the projection matrix $P^{f,i} = U^{f,i}$. This presents a trade-off between storage cost and loading overhead. If the loading overhead is a concern, one can optionally store $U^{f,i}$. Nonetheless, in scenarios where multiple task-specific adaptations are required from a single base model, PiCa

offers greater scalability: a shared set of task-agnostic $U^{f,i}$ can be pre-computed and paired with multiple sets of lightweight task-specific $B^f$, enabling efficient adaptation across diverse tasks.

# 6 CONCLUSION

In this work, we introduced PiCa, a parameter-efficient fine-tuning method that integrates gradient projection onto the principal column space of pre-trained weights with a novel weight-sharing mechanism. Our theoretical analysis establishes that column space projection provides an effective inductive bias for fine-tuning, while the addition of weight-sharing offers substantial reductions in trainable parameters without compromising performance. Through extensive experiments, we demonstrated that PiCa consistently achieves competitive or superior results compared to state-of-the-art baselines across a wide spectrum of NLP tasks (Mathematical Reasoning, Commonsense Reasoning, and Natural Language Understanding) as well as challenging vision tasks (Visual Adaptation and Subject-Driven Generation).

Taken together, our results indicate that PiCa offers a theoretically grounded and empirically validated approach to parameter-efficient adaptation of large models. We hope this work motivates further exploration of theoretically guided approaches that unify geometry-aware design with practical efficiency in fine-tuning large-scale foundation models. In future work, we aim to extend PiCa to more dynamic and practical settings such as multi-task adaptation and continual learning, where efficient and scalable fine-tuning is critical.

## ACKNOWLEDGMENT

This work was supported by the National Research Foundation of Korea (NRF) grant funded by the Korea government (MSIT) (No. RS-2024-00345809, Research on AI Robustness Against Distribution Shift in Real-World Scenarios; and No. RS-2023-00222663, Center for Optimizing Hyperscale AI Models and Platforms).

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

# APPENDIX

## A PRELIMINARIES

### A.1 NOTATION

**Notation 1.** The following notation is used throughout this paper:

- For any matrix $A \in \mathbb{R}^{m \times n}$, let $\sigma_i(A)$ denote its $i$-th largest singular value, with $\sigma_1(A) \geq \sigma_2(A) \geq \cdots \geq \sigma_{\min(m,n)}(A) \geq 0$.

- $\|A\|_F$: Frobenius norm of matrix $A$, defined as $\|A\|_F = \sqrt{\sum_{i,j} A_{ij}^2}$.

- $\|A\|_2$: Spectral norm of matrix $A$, defined as $\|A\|_2 = \sigma_1(A)$.

- $A_{ij}$: Entry at the $i$-th row and $j$-th column of matrix $A$.

- $I_k$: Identity matrix of size $k \times k$.

- $\mathrm{diag}(a_1, \ldots, a_n)$: Diagonal matrix with entries $a_1, \ldots, a_n$.

- $\sin \Theta(U_r, U_r^*)$: denotes the principal angles between the subspaces $\mathrm{range}(U_r)$ and $\mathrm{range}(U_r^*)$.

### A.2 PRELIMINARY RESULTS

**Lemma A.1** (Weyl's Inequality (Weyl, 1912)). *For $A, B \in \mathbb{R}^{m \times n}$, and all $i$,*

$$|\sigma_i(A + B) - \sigma_i(A)| \leq \|B\|_2.$$

**Lemma A.2** (Invariance of Frobenius Norm). *If $A \in \mathbb{R}^{m \times n}$, and $U, V$ are orthogonal matrices, then*

$$\|U A V^T\|_F = \|A\|_F.$$

**Lemma A.3** (Orthogonal projection is non-expansive in Frobenius norm). *Let $U_r \in \mathbb{R}^{m \times r}$ have orthonormal columns and let $\Pi_{U_r} = U_r U_r^\top$ be the orthogonal projector onto $\mathrm{range}(U_r)$. Then, for all $X \in \mathbb{R}^{m \times n}$,*

$$\|\Pi_{U_r} X\|_F \leq \|X\|_F$$

## B PROOF OF THEOREMS

**Theorem 1** (Approximation error of projection onto $U_r$). *Let $W_0 = U \Sigma V^\top \in \mathbb{R}^{m \times n}$ be the Singular Value Decomposition (SVD) of $W_0$. Suppose the fine-tuned matrix $W^* \in \mathbb{R}^{m \times n}$ has the form*

$$W^* = (UP)\Sigma^*(VQ)^\top,$$

*where:*

- *$U^* = UP$ and $V^* = VQ$ are the left and right singular vectors of $W^*$, respectively,*

- *$\Sigma^* = diag(\sigma_1(W^*), \ldots, \sigma_{\min(m,n)}(W^*))$,*

- *$P = I_m + E^P$, $Q = I_n + E^Q$, with $|E_{ij}^P| < \epsilon$, $|E_{ij}^Q| < \epsilon$.*

*Let $\Delta W = W^* - W_0$, and let $U_r \in \mathbb{R}^{m \times r}$ be the top-$r$ left singular vectors of $W_0$. Then, the approximation error incurred by projecting $\Delta W$ onto the subspace spanned by $U_r$ satisfies*

$$\left\|\Delta W - U_r U_r^\top \Delta W\right\|_F^2 \leq \sum_{i=r+1}^{\min(m,n)} \sigma_i^2(\Delta W) + \mathcal{O}(\epsilon).$$

*Proof.* We derive the inequality through a series of steps, decomposing the perturbation, analyzing the projection error, and bounding the terms using spectral and entrywise techniques.

The perturbed matrix has the form

$$W^* = U(I_m + E^P)\Sigma^*(I_n + E^Q)^\top V^\top.$$

Subtracting $W_0 = U\Sigma V^\top$ gives

$$\Delta W = U\left[(I_m + E^P)\Sigma^*(I_n + E^Q)^\top - \Sigma\right]V^\top.$$

For notational clarity, define

$$H = (I_m + E^P)\Sigma^*(I_n + E^Q)^\top - \Sigma,$$

so that $\Delta W = UHV^\top$.

Let us expand $H$ explicitly. Multiplying out terms yields

$$(I_m + E^P)\Sigma^*(I_n + E^Q)^\top = \Sigma^* + E^P\Sigma^* + \Sigma^*(E^Q)^\top + E^P\Sigma^*(E^Q)^\top.$$

Thus

$$H = D + E_1 + E_2 + E_3,$$

where

$$D = \Sigma^* - \Sigma, \quad E_1 = E^P\Sigma^*, \quad E_2 = \Sigma^*(E^Q)^\top, \quad E_3 = E^P\Sigma^*(E^Q)^\top.$$

The diagonal matrix $D$ captures the shifts in singular values: $D_{ii} = \sigma_i(W^*) - \sigma_i(W_0)$.

The error of projecting $\Delta W$ onto $U_r$ is

$$\|\Delta W - U_r U_r^\top \Delta W\|_F^2.$$

Since $\Delta W = UHV^\top$ and $U_r^\top U = [I_r\ 0]$, we can write

$$U_r U_r^\top \Delta W = U\begin{bmatrix} I_r & 0 \\ 0 & 0 \end{bmatrix} HV^\top.$$

Subtracting gives

$$\Delta W - U_r U_r^\top \Delta W = U(H - P_r H)V^\top,$$

where $P_r = \begin{bmatrix} I_r & 0 \\ 0 & 0 \end{bmatrix}$. By invariance of the Frobenius norm,

$$\|\Delta W - U_r U_r^\top \Delta W\|_F^2 = \|H - P_r H\|_F^2 = \sum_{i=r+1}^{m}\sum_{j=1}^{n} H_{ij}^2.$$

For $i > r$, each entry has the form

$$H_{ij} = D_{ij} + E_{1,ij} + E_{2,ij} + E_{3,ij}.$$

For diagonal terms ($j = i$), we have

$$H_{ii} = \sigma_i(W^*) - \sigma_i(W_0) + E_{ii}^P\sigma_i(W^*) + \sigma_i(W^*)E_{ii}^Q + \sum_k E_{ik}^P\sigma_k(W^*)E_{ik}^Q.$$

Using $|E_{ij}^P|, |E_{ij}^Q| < \epsilon$, we can bound each component:

$$|E_{1,ii}| \leq \epsilon\sigma_i(W^*), \quad |E_{2,ii}| \leq \epsilon\sigma_i(W^*), \quad |E_{3,ii}| \leq \epsilon^2 \min(m,n)\,\sigma_{\max}(W^*).$$

For off-diagonal terms ($j \neq i$), we have

$$H_{ij} = E_{ij}^P\sigma_j(W^*) + \sigma_i(W^*)E_{ji}^Q + \sum_k E_{ik}^P\sigma_k(W^*)E_{jk}^Q,$$

leading to analogous bounds

$$|E_{1,ij}| \leq \epsilon\sigma_j(W^*), \quad |E_{2,ij}| \leq \epsilon\sigma_i(W^*), \quad |E_{3,ij}| \leq \epsilon^2 \min(m,n)\,\sigma_{\max}(W^*).$$

We now square and sum these contributions. For diagonals,

$$H_{ii}^2 = (\sigma_i(W^*) - \sigma_i(W_0))^2 + 2(\sigma_i(W^*) - \sigma_i(W_0))(E_{1,ii} + E_{2,ii} + E_{3,ii}) + (E_{1,ii} + E_{2,ii} + E_{3,ii})^2.$$

Cross term is bounded using Cauchy–Schwarz, and third quadratic term is bounded by $3(E_{1,ii}^2 + E_{2,ii}^2 + E_{3,ii}^2)$. Therefore,

$$\sum_{i=r+1}^{\min(m,n)} H_{ii}^2 \leq \sum_{i=r+1}^{\min(m,n)} (\sigma_i(W^*) - \sigma_i(W_0))^2 + \epsilon C_1 + \epsilon^2 C_2.$$

where

$$C_1 = \sum_{i=r+1}^{\min(m,n)} 2|\sigma_i(W^*) - \sigma_i(W_0)|(2\sigma_i(W^*) + \epsilon \min(m,n)\sigma_{max}(W^*))$$

$$C_2 = \sum_{i=r+1}^{\min(m,n)} 3(2\sigma_i^2(W^*) + \epsilon^2 \min(m^2, n^2)\sigma_{max}^2(W^*)$$

Similar expansions apply for off-diagonal terms, where only $E_1, E_2, E_3$ contribute. For off-diagonal terms:

$$\sum_{i=r+1}^{m} \sum_{\substack{j=1 \\ j \neq i}}^{n} H_{ij}^2 = \sum_{i=r+1}^{m} \sum_{\substack{j=1 \\ j \neq i}}^{n} (E_{1,ij} + E_{2,ij} + E_{3,ij})^2 \leq \sum_{i=r+1}^{m} \sum_{\substack{j=1 \\ j \neq i}}^{n} 3(E_{1,ij}^2 + E_{2,ij}^2 + E_{3,ij}^2) \leq \epsilon^2 C_3.$$

where

$$C_3 = \sum_{i=r+1}^{m} \sum_{\substack{j=1 \\ j \neq i}}^{n} 3(\sigma_j^2(W^*) + \sigma_i^2(W^*) + \epsilon^2 \min(m^2, n^2)\sigma_{max}^2(W^*))$$

Collecting everything, the sum takes the form

$$\sum_{i=r+1}^{m} \sum_{j=1}^{n} H_{ij}^2 \leq \sum_{i=r+1}^{\min(m,n)} (\sigma_i(W^*) - \sigma_i(W_0))^2 + \epsilon C_1 + \epsilon^2 (C_2 + C_3).$$

Recall the decomposition

$$H = D + E_1 + E_2 + E_3, \qquad \Delta W = UHV^\top,$$

so that by orthogonal invariance of singular values

$$\sigma_i(\Delta W) = \sigma_i(H) \quad \text{for all } i.$$

Since $UP$ and $VQ$ are the singular-vector matrices of $W^*$, the factors $P, Q$ are orthogonal. Hence

$$D = \Sigma^* - \Sigma \quad \Rightarrow \quad \sigma_i(D) = |\sigma_i(W^*) - \sigma_i(W_0)| \quad (\forall i).$$

Let $E_{\text{tot}} := E_1 + E_2 + E_3$. By Weyl's inequality applied to $H = D + E_{\text{tot}}$,

$$|\sigma_i(H) - \sigma_i(D)| = |\sigma_i(\Delta W) - |\sigma_i(W^*) - \sigma_i(W_0)|| \leq \|E_{\text{tot}}\|_2.$$

We now bound $\|E_{\text{tot}}\|_2$ piecewise. Using submultiplicativity and $\|E^P\|_2 \leq \|E^P\|_F \leq \sqrt{mn}\,\epsilon$ (and similarly for $E^Q$), we get

$$\|E_1\|_2 = \|E^P \Sigma^*\|_2 \leq \|E^P\|_2 \|\Sigma^*\|_2 \leq \sqrt{mn}\,\epsilon\,\sigma_{\max}(W^*),$$

$$\|E_2\|_2 = \|\Sigma^* (E^Q)^\top\|_2 \leq \|\Sigma^*\|_2 \|E^Q\|_2 \leq \sqrt{mn}\,\epsilon\,\sigma_{\max}(W^*),$$

$$\|E_3\|_2 = \|E^P \Sigma^* (E^Q)^\top\|_2 \leq \|E^P\|_2 \|\Sigma^*\|_2 \|E^Q\|_2 \leq mn\,\epsilon^2\,\sigma_{\max}(W^*).$$

Therefore

$$\|E_{\text{tot}}\|_2 \leq 2\sqrt{mn}\,\epsilon\,\sigma_{\max}(W^*) + mn\,\epsilon^2\,\sigma_{\max}(W^*).$$

Define

$$\delta_i \; := \; \sigma_i(\Delta W) \; - \; \big|\sigma_i(W^*) - \sigma_i(W_0)\big|, \qquad |\delta_i| \; \leq \; \|E_{\text{tot}}\|_2.$$

Then

$$\big|\sigma_i(W^*) - \sigma_i(W_0)\big| \; = \; \sigma_i(\Delta W) - \delta_i,$$

and squaring gives

$$\big(\sigma_i(W^*) - \sigma_i(W_0)\big)^2 \; = \; \big(\sigma_i(\Delta W) - \delta_i\big)^2 \; \leq \; \sigma_i^2(\Delta W) \; + \; 2\,\sigma_i(\Delta W)\,\|E_{\text{tot}}\|_2 \; + \; \|E_{\text{tot}}\|_2^2.$$

Let $\ell := \min(m, n)$. Summing for $i = r+1, \ldots, \ell$,

$$\sum_{i=r+1}^{\ell} \big(\sigma_i(W^*) - \sigma_i(W_0)\big)^2 \; \leq \; \sum_{i=r+1}^{\ell} \sigma_i^2(\Delta W) \; + \; 2\,\|E_{\text{tot}}\|_2 \sum_{i=r+1}^{\ell} \sigma_i(\Delta W) \; + \; (\ell - r)\,\|E_{\text{tot}}\|_2^2.$$

With the bound on $\|E_{\text{tot}}\|_2$ just obtained, this can be written as

$$\sum_{i=r+1}^{\ell} \big(\sigma_i(W^*) - \sigma_i(W_0)\big)^2 \; \leq \; \sum_{i=r+1}^{\ell} \sigma_i^2(\Delta W) \; + \; \epsilon\, C_4 \; + \; \epsilon^2\, C_5,$$

where

$$C_4 \; = \; 2\Big(2\sqrt{mn}\,\sigma_{\max}(W^*) + mn\,\epsilon\,\sigma_{\max}(W^*)\Big) \sum_{i=r+1}^{\ell} \sigma_i(\Delta W),$$

$$C_5 \; = \; (\ell - r)\Big(2\sqrt{mn}\,\sigma_{\max}(W^*) + mn\,\epsilon\,\sigma_{\max}(W^*)\Big)^2.$$

Finally, recalling the earlier analysis, we finally combine the bounds to obtain

$$\|\Delta W - U_r U_r^\top \Delta W\|_F^2 \leq \sum_{i=r+1}^{\min(m,n)} \sigma_i^2(\Delta W) + \epsilon C_1 + \epsilon^2 C_2 + \epsilon^2 C_3 + \epsilon C_4 + \epsilon^2 C_5$$

$$= \sum_{i=r+1}^{\min(m,n)} \sigma_i^2(\Delta W) + \epsilon C$$

where

$$C = (C_1 + \epsilon C_2 + \epsilon C_3 + C_4 + \epsilon C_5)$$

$\square$

**Theorem 2** (Sequential projection approximates accumulated projection). *Let $\ell : \mathbb{R}^{m \times n} \to \mathbb{R}$ be L-smooth with $\|\nabla\ell(W)\|_F \leq G$. Define the unprojected gradient descent path*

$$Z_{t+1} = Z_t - \eta \nabla\ell(Z_t).$$

*Let the* accumulated-projection *iterate be*

$$W_T = W_0 - \eta \, \Pi_{U_r}\Big(\sum_{t=0}^{T-1} \nabla\ell(Z_t)\Big),$$

*and the* sequential-projection *iterates*

$$P_{t+1} = P_t - \eta \, \Pi_{U_r} \nabla\ell(P_t), \qquad P_0 = W_0,$$

*where $\Pi_{U_r} = U_r U_r^\top$ is the fixed rank-r projector.*

*Then, for any $T$, the difference satisfies*

$$\|W_T - P_T\|_F \; \leq \; \frac{\eta^2}{2} \, LG\,T(T-1) + O((\eta L T)^3).$$

*Proof.* We now prove that the sequentially projected iterates closely approximate the delayed projection iterate when both use the same fixed projector $\Pi_{U_r} = U_r U_r^\top$. Throughout we work with the Frobenius norm, and recall from Lemma A.3 that $\Pi_{U_r}$ is non-expansive in $\|\cdot\|_F$.

The delayed projection iterate is defined by

$$W_T^{\text{delayed}} = W_0 - \eta\,\Pi_{U_r}\Big(\sum_{t=0}^{T-1}\nabla\ell(Z_t)\Big), \qquad Z_{t+1} = Z_t - \eta\nabla\ell(Z_t).$$

The sequentially projected iterates follow

$$P_{t+1} = P_t - \eta\,\Pi_{U_r}\nabla\ell(P_t), \qquad P_0 = W_0.$$

Subtracting the two update rules yields

$$P_T - W_T^{\text{delayed}} = -\eta\sum_{t=0}^{T-1}\Pi_{U_r}\big(\nabla\ell(P_t) - \nabla\ell(Z_t)\big).$$

Taking Frobenius norms and using $\|\Pi_{U_r}\|_{F\to F} \le 1$,

$$\|P_T - W_T^{\text{delayed}}\|_F \ \le\ \eta\sum_{t=0}^{T-1}\|\nabla\ell(P_t) - \nabla\ell(Z_t)\|_F.$$

By Definition 1, $\ell$ is $L$-smooth w.r.t. $\|\cdot\|_F$, so the gradient is $L$-Lipschitz:

$$\|\nabla\ell(P_t) - \nabla\ell(Z_t)\|_F \ \le\ L\|P_t - Z_t\|_F.$$

Denoting $D_t = \|P_t - Z_t\|_F$, we obtain

$$\|P_T - W_T^{\text{delayed}}\|_F \ \le\ \eta L\sum_{t=0}^{T-1}D_t.$$

To bound $D_t$, expand one step of the deviation:

$$\begin{aligned}
D_{t+1} &= \|P_{t+1} - Z_{t+1}\|_F \\
&= \|P_t - \eta\Pi_{U_r}\nabla\ell(P_t)\ -\ (Z_t - \eta\nabla\ell(Z_t))\|_F \\
&= \|P_t - Z_t - \eta(\Pi_{U_r}\nabla\ell(P_t) - \nabla\ell(Z_t))\|_F.
\end{aligned}$$

Applying the triangle inequality and splitting terms,

$$D_{t+1} \le D_t + \eta\,\|\Pi_{U_r}(\nabla\ell(P_t) - \nabla\ell(Z_t))\|_F + \eta\,\|(I - \Pi_{U_r})\nabla\ell(Z_t)\|_F.$$

For the first term, by non-expansiveness of $\Pi_{U_r}$ and $L$-smoothness,

$$\|\Pi_{U_r}(\nabla\ell(P_t) - \nabla\ell(Z_t))\|_F \ \le\ \|\nabla\ell(P_t) - \nabla\ell(Z_t)\|_F \ \le\ LD_t.$$

For the second term, since $\|\nabla\ell(Z_t)\|_F \le G$ by assumption,

$$\|(I - \Pi_{U_r})\nabla\ell(Z_t)\|_F \ \le\ \|\nabla\ell(Z_t)\|_F \ \le\ G.$$

Hence the recurrence is

$$D_{t+1} \ \le\ (1+\eta L)D_t + \eta G.$$

With $D_0 = 0$, a standard unrolling argument gives

$$D_t \ \le\ \frac{G}{L}\big((1+\eta L)^t - 1\big) \ \le\ \frac{G}{L}\big(e^{\eta L t} - 1\big).$$

Plugging back into Step 2,

$$\|P_T - W_T^{\text{delayed}}\|_F \ \le\ \eta L\sum_{t=0}^{T-1}D_t \ \le\ \eta G\sum_{t=0}^{T-1}(e^{\eta L t} - 1).$$

For small $\eta LT$, we use the second-order Taylor expansion of the exponential:

$$e^x - 1 = x + \frac{x^2}{2} + O(x^3) \quad \text{as } x \to 0.$$

Applying this with $x = \eta Lt$ yields

$$e^{\eta Lt} - 1 \;=\; \eta Lt + \tfrac{1}{2}(\eta Lt)^2 + O\big((\eta Lt)^3\big),$$

and hence

$$\eta L \sum_{t=0}^{T-1} D_t \;\le\; \eta G \sum_{t=0}^{T-1} (e^{\eta Lt} - 1) \;=\; \frac{\eta^2 LG}{2} T(T-1) \;+\; O\big((\eta LT)^3\big).$$

Combining all estimates, we conclude

$$\|W_T - P_T\|_F \;\le\; \frac{\eta^2 LG}{2} T(T-1) \;+\; O\big((\eta LT)^3\big),$$

which shows that the sequential projection scheme faithfully tracks the delayed projection up to higher-order error in the learning rate and horizon.

$\square$

## C  IMPLEMENTATION DETAILS AND ADDITIONAL EXPERIMENTS

To ensure a direct and unbiased comparison with existing baseline methods, we adopted the same experimental setup as outlined in SVFT (Lingam et al., 2024) for NLP tasks. For consistency, all baseline results in NLP tasks were also sourced from (Lingam et al., 2024), enabling a fair evaluation of our method's performance. For vision tasks, we follow Dosovitskiy et al. (2021) and Cho et al. (2024).

For Mathematical Reasoning tasks, we reproduced Full Fine-Tuning (Full FT) experiments for Gemma-7B and LLaMA-3-8B using smaller learning rates. While previous results were reported in (Lingam et al., 2024), we found that with sufficiently low learning rates (1e-6/5e-6), Full FT achieves higher accuracy than PEFT methods. We suspect SVFT used higher learning rates (1e-5/5e-5) for Full FT. The comparison is summarized in Table 6.

Table 6: Full Fine-Tuning (Full FT) accuracy on GSM8K and MATH.

| Setting | Gemma-7B | | LLaMA-3-8B | |
|---|---|---|---|---|
| | GSM8K | MATH | GSM8K | MATH |
| Reported in SVFT | 74.67 | 25.70 | 64.13 | 16.24 |
| Our Reproduction (Low LR) | **78.09** | **30.98** | **76.57** | **26.12** |

### C.1  IMPLEMENTATION DETAILS

**Mathematical Reasoning**  Table 7 presents the hyperparameter configurations employed for these experiments. For the Gemma model family, PiCa is applied to the $Q, K, V, U, D$ matrices, while for the LLaMA-3-8B model, the $Q, K, V, U, D, O, G$ matrices are targeted. The experimental codebase and evaluation procedures are adapted from `https://github.com/VijayLingam95/SVFT.git`, and the fine-tuning dataset are sourced from `https://huggingface.co/datasets/meta-math/MetaMathQA-40K`.

**Commonsense Reasoning**  We follow the setting outlined in prior work (Lingam et al., 2024), fine-tuning on 15K examples. The hyperparameter configurations for these experiments are detailed in  Table 8. We utilize the same set of matrices as in the Mathematical Reasoning tasks. The codebase, including training and evaluation data, is sourced from `https://github.com/VijayLingam95/SVFT.git`.

Table 7: Hyperparameter setup used for fine-tuning on MetaMathQA-40K.

| Hyperparameter | Gemma-2B | | Gemma-7B | | LLaMA-3-8B | |
|---|---|---|---|---|---|---|
| Optimizer | | | AdamW | | | |
| Warmup Ratio | | | 0.1 | | | |
| LR Schedule | | | Cosine | | | |
| Max Seq. Len. | | | 512 | | | |
| # Epochs | | | 2 | | | |
| Batch Size | | | 64 | | | |
| Rank | 32 | 256 | 16 | 256 | 32 | 256 |
| Learning Rate | 1E-03 | 9E-04 | 1E-04 | 5E-05 | 2E-04 | 2E-04 |

Table 8: Hyperparameter setup used for fine-tuning on commonsense-15K.

| Hyperparameter | Gemma-2B | | Gemma-7B | |
|---|---|---|---|---|
| Optimizer | | | AdamW | |
| Warmup Steps | | | 100 | |
| LR Schedule | | | Linear | |
| Max Seq. Len. | | | 512 | |
| # Epochs | | | 3 | |
| Batch Size | | | 64 | |
| Rank | 32 | 256 | 16 | 128 |
| Learning Rate | 1E-03 | 9E-04 | 3E-04 | 8E-05 |

**Natural Language Understanding** We fine-tune DeBERTaV3$_{base}$ (He et al., 2023), applying PiCa to all linear layers within each transformer block. We constrain hyperparameter optimization to moderate adjustments of the learning rate and the number of training epochs. For rigorous comparison, we employ identical model sequence lengths to those reported by (Lingam et al., 2024; Liu et al., 2024b). The precise hyperparameter settings utilized in these experiments are specified in Table 9.

Table 9: Hyperparameter setup used for DeBERTaV3$_{base}$ on the GLUE benchmark.

| Method | Dataset | MNLI | SST-2 | MRPC | CoLA | QNLI | QQP | RTE | STS-B |
|---|---|---|---|---|---|---|---|---|---|
| | Optimizer | | | | AdamW | | | | |
| | Warmup Ratio | | | | 0.1 | | | | |
| | LR Schedule | | | | Linear | | | | |
| | Batch Size | | | | 32 | | | | |
| | Max Seq. Len. | 256 | 128 | 320 | 64 | 512 | 320 | 320 | 128 |
| PiCa$_{r=16}$ | Learning Rate | 3E-04 | 1E-03 | 2E-03 | 8E-4 | 3E-04 | 1E-04 | 1E-03 | 3E-03 |
| | # Epochs | 5 | 7 | 35 | 50 | 5 | 15 | 40 | 15 |

**Vision Experiments** For vision adaptation tasks, we fine-tune ViT-B/16 (Dosovitskiy et al., 2021) by updating all linear layers within each transformer block, using a learning rate of 0.004 for PiCa and LoRA, 0.005 for DoRA, and 0.05 for VeRA and SVFT. For all methods, the classifier learning rate is fixed at 0.005. Fine-tuning is conducted for 10 epochs, and the checkpoint from the best validation epoch is used for testing. The same hyperparameter configurations are applied across all 19 datasets of VTAB-1K (Zhai et al., 2020). For subject-driven generation tasks, we follow training and evaluation protocols of previous works (Lingam et al., 2024; Cho et al., 2024). We use a learning rate of 0.0001 for LoRA and DoRA, 0.0005 for PiCa, 0.001 for SVFT, and 0.005 for VeRA. Other settings remain the same with Cho et al. (2024).

## C.2 Evidence from Large-scale Models.

While Fig. 2 provides visual evidence of subspace alignment in moderate-scale settings, here we empirically validate the assumptions underlying Theorem 1 on a larger model. Specifically, we analyze LLaMA3-8B fine-tuned on Commonsense Reasoning benchmarks.

For each pair of pre-trained and fine-tuned weight matrices, we computed the cosine similarity between their singular vectors and defined *Diagonal Similarity* as the average of the diagonal entries of the similarity matrix, aggregated across layers of each module (query, key, and value). The consistently high Diagonal Similarity values reported in Table 10 demonstrate that the leading singular subspaces remain well aligned after fine-tuning, thus supporting the subspace stability assumption of Theorem 1.

We also extend the analysis of Fig. 2 by reporting the averaged entries of $E^P$ and $E^Q$ across layers. As shown in Table 10, these values are tightly concentrated around zero, empirically confirming that the additional $\mathcal{O}(\epsilon)$ term in Theorem 1 is negligible in practice.

Table 10: Empirical validation of Theorem 1 assumptions on LLaMA3-8B fine-tuned for Commonsense Reasoning. Diagonal Similarity measures alignment of singular vectors between pre-trained and fine-tuned weights. The averaged values of $E_{ij}^P$ and $E_{ij}^Q$ are tightly concentrated near zero, confirming that the $\mathcal{O}(\epsilon)$ term is negligible.

| Layer | Diagonal Similarity | $E_{ij}^P$ | $E_{ij}^Q$ |
|-------|--------------------|-----------|-----------|
| Query | $0.927 \pm 0.047$ | $-2.44e-4 \pm 4.27e-6$ | $-2.44e-4 \pm 4.25e-6$ |
| Key   | $0.998 \pm 0.003$ | $-9.66e-4 \pm 3.76e-5$ | $-9.66e-4 \pm 3.76e-5$ |
| Value | $0.972 \pm 0.011$ | $-9.69e-4 \pm 2.76e-5$ | $-9.66e-4 \pm 2.76e-5$ |

## C.3 Commonsense Reasoning with Gemma-2B

We evaluate PiCa on commonsense reasoning tasks with Gemma-2B. The results are presented in Table 11. PiCa achieves the highest average performance across both high- and low-rank settings, outperforming the second-best method by approximately 2–3 percentage points.

Table 11: Performance on Commonsense Reasoning benchmarks using Gemma-2B. #Params refers to the number of trainable parameters. The best and second-best PEFT methods are highlighted in **bold** and underlined text, respectively. PiCa achieves state-of-the-art average performance across both high- and low-rank settings, outperforming the second-best method by up to 3 percentage points.

| Method | #Params | BoolQ | PIQA | SIQA | HS | WG | ARC-e | ARC-c | OBQA | Avg. |
|--------|---------|-------|------|------|-----|-----|-------|-------|------|------|
| Full-FT | 2.5B | 63.57 | 74.10 | 65.86 | 70.00 | 61.95 | 75.36 | 59.72 | 69.00 | 67.45 |
| BOFT$_{m=2}^{b=8}$ | 1.22M | 59.23 | 63.65 | 47.90 | 29.93 | 50.35 | 59.04 | 42.66 | 41.00 | 49.22 |
| VeRA$_{r=2048}$ | 0.66M | 62.11 | 64.31 | 49.18 | 32.00 | 50.74 | 58.08 | 42.83 | 42.60 | 50.23 |
| LoRA$_{r=1}$ | 0.82M | 62.20 | 69.31 | 56.24 | 32.47 | **51.53** | 69.52 | 48.80 | 56.40 | 55.81 |
| DoRA$_{r=1}$ | 1.19M | 62.17 | 68.77 | 55.93 | 32.95 | 51.22 | 68.81 | 48.72 | 55.60 | 55.52 |
| SVFT$_P$ | 0.19M | **62.26** | 70.18 | 56.70 | 32.47 | 47.04 | 69.31 | 50.08 | 58.40 | 55.81 |
| PiCa$_{r=32}$ | 0.67M | 62.11 | **71.76** | **60.13** | **36.49** | 50.59 | **73.74** | **52.56** | **63.20** | **58.82** |
| LoRA$_{r=32}$ | 26.2M | 63.11 | 73.44 | 63.20 | 47.79 | 52.95 | 74.78 | 57.16 | 67.00 | 62.43 |
| DoRA$_{r=16}$ | 13.5M | 62.87 | 73.93 | **65.34** | 53.16 | 55.51 | 76.43 | **59.55** | **68.40** | 64.40 |
| SVFT$_B^{d=16}$ | 6.35M | 63.42 | 73.72 | 63.86 | 71.21 | 59.58 | 73.69 | 54.77 | 66.60 | 65.86 |
| PiCa $_{r=256}$ | 5.37M | **63.91** | **75.57** | 64.38 | **71.75** | **60.62** | **77.44** | 58.70 | **68.40** | **67.60** |

## C.4 Resource and Efficiency Analysis

We present a comparative analysis of training memory usage between PiCa and the state-of-the-art baseline SVFT. Training memory is measured by peak GPU memory usage during fine-tuning. As

Table 12: Trainable parameters and training memory consumption for different parameter-efficient fine-tuning methods on Gemma-7B.

| Method | LoRA$_{r=32}$ | DoRA$_{r=16}$ | BOFT$_{m=2}^{b=8}$ | VeRA$_{r=1024}$ | SVFT$^P$ | SVFT$_{d=16}^R$ | PiCa$_{r=128}$ |
|---|---|---|---|---|---|---|---|
| # Params | 68.8M | 35.5M | 2.90M | 0.43M | 0.43M | 19.8M | 5.11M |
| Memory | 52,764 | 55,334 | 65,423 | 55,980 | 70,176 | 71,421 | 52,614 |

shown in Table 12, although both SVFT$^P$ and SVFT$_{d=16}^R$ substantially reduce the number of trainable parameters, they require up to 36% higher GPU memory consumption compared to PiCa$_{r=128}$.

This overhead arises from SVFT's factorization of weight updates as $\Delta W = UMV^\top$, where $U \in \mathbb{R}^{m \times m}$ and $V \in \mathbb{R}^{n \times n}$ denote the singular vectors of the pre-trained weight matrix $W_0 \in \mathbb{R}^{m \times n}$. Although $U$ and $V$ are not trainable, they must be retained throughout fine-tuning, leading to significant memory overhead. These results demonstrate that PiCa offers a more memory-efficient alternative, particularly in resource-constrained environments where memory I/O constitutes a critical bottleneck.

## C.5 COMPARISON UNDER WEIGHT-SHARING

We compare PiCa with PEFT baselines under the same weight-sharing scheme. For SVFT, we report both Random and Banded variants with sharing. As shown in Table 13, PiCa achieves the best performance while using fewer parameters.

Table 13: Results with weight-sharing on GSM-8K and MATH using Gemma-2B.

| Method | #Params | GSM-8K | MATH |
|---|---|---|---|
| LoRA (r=32, $B$ shared) | 14.83M | 44.28 | 15.08 |
| DoRA (r=16, $B$ shared) | 7.79M | 41.02 | 14.92 |
| SVFT_R w/ sharing ($d = 280$) | 5.48M | 50.49 | 15.86 |
| SVFT_B w/ sharing ($d = 280$) | 5.48M | 50.34 | 16.18 |
| PiCa (r=256) | 5.37M | **52.77** | **16.36** |

## D LLM USAGE

We used large language models only for minor tasks such as spell-checking, grammar correction, and formatting.

## E REPRODUCIBILITY STATEMENT

We have made extensive efforts to ensure the reproducibility of our work. All models, datasets, training protocols, and hyperparameters required to reproduce our experimental results are described in detail in Section 4 and Appendix C.

