# OpenReview forum: "PiCa: Parameter-Efficient Fine-Tuning with Column Space Projection"
_ICLR.cc/2026/Conference — ICLR 2026 Poster_

### Official Review · Reviewer_XBE4 · 2025-10-30

**Soundness:** 3
**Presentation:** 2
**Contribution:** 2
**Rating:** 4
**Confidence:** 4

**Summary:**

This paper introduces PiCa, a novel and theoretically grounded method for parameter-efficient fine-tuning (PEFT) of large foundation models. The core idea is to project gradients onto the principal column space of pre-trained weights, a space empirically and theoretically shown to preserve fine-tuning directions with minimal update error. To further reduce trainable parameters, PiCa integrates a weight-sharing mechanism, where shared compact parameters are used across layers within the same functional group. The proposed method is evaluated across a wide range of NLP (Mathematical Reasoning, Commonsense Reasoning, GLUE) and vision tasks (VTAB-1K, DreamBooth). PiCa outperforms state-of-the-art PEFT baselines, including LoRA, DoRA, SVFT, and VeRA, under equal or lower parameter budgets.

**Strengths:**

- **Innovative method with practical benefits:** The combination of gradient projection and parameter sharing is novel and effective. The layer-specific projection matrices allow for structural adaptivity, while shared compact parameters reduce redundancy, enabling PiCa to use up to 13× fewer parameters than LoRA without sacrificing performance.

- **Comprehensive experiments:** PiCa is extensively evaluated across three NLP subfields and two vision domains, including strong baselines. The method shows superior average performance across tasks such as GLUE, GSM-8K, and VTAB-1K, highlighting its general applicability beyond a single modality.

**Weaknesses:**

- **Inference-time overhead not deeply addressed:** PiCa requires computing (or storing) the top singular vectors of each pre-trained layer for projection. Although the authors mention this as a minor load-time overhead, more discussion or benchmarking of real-world inference-time latency/storage trade-offs would be helpful.

- **Lack of comparison on long-context or sequence-heavy tasks:** While the NLP evaluations are diverse, no experiments are shown on long-context reasoning, e.g., document-level QA or summarization. Since PiCa modifies fine-tuning dynamics, it remains unclear how it scales in such scenarios.

**Questions:**

- **How sensitive is PiCa to the choice of projection rank $r$?** Although some rank settings are explored, it would be helpful to know if PiCa is **robust across a range of $r$**, especially in lower-resource settings. Could adaptive rank selection (e.g., via energy thresholding) be incorporated?

- **Can PiCa be combined with LoRA or orthogonal fine-tuning?** Given its theoretical grounding, could PiCa act as a drop-in enhancement to existing PEFT layers like LoRA by refining the directionality of updates?

- Can the authors comment on scalability in terms of actual wall-clock latency or GPU memory?

- The shared representation $\theta_f$ and structured projection matrix offer intriguing possibilities for compositional fine-tuning. Could this architecture be extended to **task-conditioned adaptation**?

---

> ### Author Response · Authors · 2025-11-23
>
> We thank the reviewer for their valuable feedback. We address the weaknesses (**W**) and questions (**Q**) raised by the reviewer below.
>
> &nbsp;
>
> **W2** : We agree with the reviewer that evaluating PiCa on long-context or sequence-heavy tasks can be valuable. However, training with long-context tasks requires extreme computational demands, and long-context settings often exceed the base model’s context window or require specialized mechanisms that go beyond the typical scope of general PEFT literature. We believe extending PiCa to long-context fine-tuning can be an interesting direction for future study.
>
> &nbsp;
>
> **Q1**: In Fig. 3b of our paper, we analyze PiCa under different rank settings in lower-resource settings (~10M trainable parameters), and PiCa consistently outperforms LoRA under comparable parameter budgets. Adaptive rank selection is also feasible in principle. For example, we can define an importance score and prune less important basis vectors during training. We believe extending PiCa with adaptive rank selection can be a promising direction for future research.
>
> &nbsp;
>
> **Q2**: PiCa, LoRA, and orthogonal fine-tuning each introduce their own $\Delta W$ parameterization during training, so PiCa is not designed to be layered on top of these methods as a drop-in enhancement. While the approaches are conceptually related, combining them would require a different update mechanism, which we leave as interesting future work.
>
> &nbsp;
>
> **Q3, W1**: Following your suggestion, we conducted additional experiments on actual wall-clock latency or GPU memory with Gemma-2B. As shown in the table below, PiCa achieves higher performance than SVFT while using fewer parameters and less peak GPU memory during training. In terms of inference, although both PiCa and SVFT require an SVD recomputation before fusing with the base model, this cost occurs only once. After fusion, there is no additional overhead, and inference time is identical to other PEFT methods.
>
>
> | Method            | #Params | Training Memory   | GSM-8K | MATH  |
> |-------------------|---------|----------|--------|-------|
> | SVFT_R (d=16)     | 6.35M   | 20.68GB  | 50.03  | 15.56 |
> | PiCa (r=256)      | 5.37M   | 16.67GB  | 52.77  | 16.36 |
>
> | Method             | Total (mean) | Total (std) | Base Load | Adapter Load | SVD Time | Merge Time |
> |--------------------|--------------|-------------|-----------|--------------|----------|------------|
> | SVFT_R (d=16)      | 32.22 s      | 0.20 s      | 1.92 s    | 0.06 s       | 29.95 s  | 0.29 s     |
> | PiCa (r=256)       | 16.84 s      | 0.04 s      | 1.81 s    | 0.01 s       | 14.97 s  | 0.06 s
>
> &nbsp;
>
> **Q4**: Yes, this architecture could be extended to task-conditioned adaptation. Making $\theta_f$​ task-specific would allow a task-conditioned variant of PiCa. We believe extending PiCa with compositional fine-tuning can be a promising direction for future research.

---

### Official Review · Reviewer_8LsH · 2025-11-01

**Soundness:** 2
**Presentation:** 3
**Contribution:** 2
**Rating:** 4
**Confidence:** 4

**Summary:**

This paper introduces PiCa, a theoretically grounded PEFT method that exploits of the spectral geometry of the pre-trained weights. PiCa projects gradient updates onto the principal column space (top-rank singular vectors) of each pretrained weight matrix, supported by theoretical analysis showing this projection yields a near-optimal low-rank update. To further reduce trainable parameters, PiCa also shares the projected update matrix across layers of the same type. Experiments across multiple NLP and some vision tasks demonstrate consistent state-of-the-art performance compared to previous SoTA PEFT baselines with fewer trainable parameters.

**Strengths:**

- By projecting gradients onto the top singular vectors’ subspace, PiCa leverages a mathematically principled inductive bias. The authors prove that this yields a near-optimal low-rank update (minimizing Frobenius error) up to small residual terms.

- Extensive experiments validate that PiCa achieves competitive or superior performance while using fewer parameters than baselines.

**Weaknesses:**

- PiCa’s theoretical guarantee (Theorem 1) assumes that the transformation from pre-trained to fine-tuned singular vectors is almost identity, i.e. fine-tuning doesn’t drastically alter the singular directions. While empirically supported for conducted experiments, this might not universally hold. If a downstream task requires introducing entirely new features or directions that were not significant in the pre-trained model, limiting updates to the original top-r singular space could be restrictive. In cases where the task’s optimal solution exists outside the column space of the pretrained weights (e.g., OOD tasks), PiCa would then incur an approximation error equal to the omitted singular components.

- Weight-sharing theoretically reduces flexibility; broader ablations on when sharing hurts would strengthen claims. Running PiCa on some OOD downstream tasks, or on less capable base LLM models would offer more insights.

**Questions:**

1. Could you apply the same weight-sharing scheme to SVFT so both methods use a comparable trainable-parameter budget? This would isolate the effect of PiCa’s rank-r projection versus SVFT’s full-rank update.

2.  Have you assessed robustness to decoding hyperparameters (e.g., temperature, top-p) and reported mean ± std over multiple seeds? Re-running with varied sampling settings would strengthen claims of statistical reliability.

---

> ### Author Response · Authors · 2025-11-23
>
> We thank the reviewer for their valuable feedback. We address the weaknesses (**W**) and questions (**Q**) raised by the reviewer below.
>
> &nbsp;
>
> **W1**: **While hypothetically there may exist scenarios in which our method might not work well if the downstream task requires directions outside the distribution of pre-trained datasets, such cases are rare for large foundation models given their broad and generalizable pretraining datasets.** Aside from extreme scenarios (e.g., synthetic or minority languages), we are not aware of open-source datasets that reflect such conditions. Within the scope of typical NLP tasks, our empirical results on different benchmark dataset and tasks suggest that PiCa performs reliably.
>
> &nbsp;
>
> **W2**: **To compensate the reduced flexibility from weight sharing, we set the rank of the shared $\theta_f$ higher so that the shared weights retain sufficient capacity.** The ablation study is presented in Fig. 3(b). The results show that applying weight sharing while keeping the rank fixed at 32 significantly reduces the number of parameters but also results in some performance drop. However, when the rank of shared weights is increased to 256, PiCa achieves better performance while still using fewer parameters. **Regarding the less-capable/small models, our paper already includes experimental results**: DeBERTaV3-base (180M) with 8 different tasks of GLUE, and ViT-B/16 (87M) with 19 different tasks of VTAB-1K. The results show that PiCa can still maintain competitive performance using significantly fewer parameters.
>
>
> &nbsp;
>
> **Q1**: **Following the reviewer’s suggestion, we applied the same sharing scheme to SVFT** so that both methods use a comparable trainable-parameter budget. The results in the table below show that PiCa’s projection contributes to improved performance compared to SVFT.
>
>
> | Method             | #Params | GSM-8K | MATH  |
> |--------------------|---------|--------|-------|
> | PiCa (r=256)       | 5.37M   | 52.77  | 16.36 |
> | SVFT_R w/ sharing (d=280)     | 5.48M   | 50.49  | 15.86 |
>
> &nbsp;
>
> **Q2**: **Following your suggestion, we additionally evaluated PiCa and SVFT under a range of decoding hyperparameters** on Gemma-2B, reporting mean ± std over five random seeds. The results in the table below show that PiCa achieves better performance than SVFT across configurations.
>
> | Temp | Top-p | Method | #Params | GSM8K | MATH |
> |------|-------|---------|---------|--------|-------|
> | 0.1  | 1     | PiCa | 5.37M  | 52.43 ± 0.34 | 16.40 ± 0.11 |
> |      |       | SVFT | 6.35M  | 49.19 ± 0.94 | 15.98 ± 0.12 |
> | 0.1  | 0.95  | PiCa | 5.37M  | 52.55 ± 0.15 | 16.33 ± 0.24 |
> |      |       | SVFT | 6.35M  | 49.23 ± 0.82 | 15.93 ± 0.08 |
> | 0.1  | 0.9   | PiCa | 5.37M  | 52.74 ± 0.11 | 16.46 ± 0.15 |
> |      |       | SVFT | 6.35M  | 49.33 ± 0.72 | 15.90 ± 0.10 |
> | 0.2  | 1     | PiCa | 5.37M  | 51.86 ± 0.65 | 16.74 ± 0.34 |
> |      |       | SVFT | 6.35M  | 49.43 ± 1.24 | 15.91 ± 0.37 |
> | 0.2  | 0.95  | PiCa | 5.37M  | 51.93 ± 0.71 | 16.63 ± 0.34 |
> |      |       | SVFT | 6.35M  | 49.69 ± 1.15 | 15.82 ± 0.32 |
> | 0.2  | 0.9   | PiCa | 5.37M  | 51.80 ± 0.65 | 16.52 ± 0.24 |
> |      |       | SVFT | 6.35M  | 49.45 ± 0.85 | 15.86 ± 0.18 |

---

### Official Review · Reviewer_UoTk · 2025-11-03

**Soundness:** 3
**Presentation:** 2
**Contribution:** 2
**Rating:** 4
**Confidence:** 5

**Summary:**

This paper proposes a PEFT method which involves, for any given weight matrix, using the singular vector spaces of that matrix to determine which parameters to update. In particular, if a matrix W has SVD given by USV', and the top-r components are U_r and V_r, this paper proposes a low-rank update of the form U_r P V_r' - where P is a small square matrix of their trainable parameters.

Further, to decrease parameter count, they do weight-tying of these P across groups of layers.

They show improvements over a few other methods that do similar things for PEFT.

**Strengths:**

The method is clearly written. While not exactly theory, there is mathematical grounding of their method presented. They have compared against the correct set of competing methods.

**Weaknesses:**

The main issue with this paper is that it is a special case of the SVFT method, with weight tying. In particular (see the notation above in the "Summary" of this review), in SVFT the update is of the form U Q V' where Q is a matrix with <some> pre-determined sparsity pattern. The only difference in this paper is that this sparsity pattern is fixed to be the r x r "top left" square corresponding to the main singular values.

The other innovation is weight tying across layers, a standard parameter-saving technique. Indeed it is likely the gains over SVFT arise mainly from weight-tying, and not some fundamental algebraic structural difference of the update. A fairer comparison would be with weight-tying applied to other methods as well.

**Questions:**

What does specifically does Theorem 2 add to the method from an algorithmic perspective? It is a pretty standard derivation from Lipschitz analysis (which, also does not hold for these models which have ReLUs in them).

How is it that PEFT methods beat fullFT even in accuracy for the 7B and 8B models (Table 2)? Was the FullFT done properly?

---

> ### Author Response · Authors · 2025-11-23
>
> We thank the reviewer for their valuable feedback. We address the weaknesses (**W**) and questions (**Q**) raised by the reviewer below.
>
> &nbsp;
>
> **W1**: **The actual formulation and implementation of PiCa and SVFT differ in several important ways.** First, in PiCa, the learnable parameter is the $\theta_f \in \mathbb{R}^{r \times n}$ which is not equivalent to  $Q\in \mathbb{R}^{r \times r}$  in the $UQV'$ parameterization. Second, while SVFT selects its sparsity patterns randomly and heuristically, since it does not know which singular directions of $W_0$ are most effective, PiCa is motivated by Theorem 1, which provides a principled reason for using the top-$r$ subspace $U_r$ for constructing the update. Lastly, from an implementation perspective, SVFT must keep both projection matrices $U \in \mathbb{R}^{m \times m}$ and $V' \in \mathbb{R}^{n \times n}$, which can be memory-intensive for large models. In contrast, PiCa only keeps the compact projection $U_r \in \mathbb{R}^{m \times r}$, resulting in a substantially lighter memory footprint and a more scalable update representation. The comparison of their peak GPU memory usage is presented in the table below.
>
> | Method            | #Params | Training Memory   | GSM-8K | MATH  |
> |-------------------|---------|----------|--------|-------|
> | SVFT_R (d=16)     | 6.35M   | 20.68GB  | 50.03  | 15.56 |
> | PiCa (r=256)      | 5.37M   | 16.67GB  | 52.77  | 16.36 |
>
>
> &nbsp;
>
> **W2**: **Following the reviewer’s suggestion, we applied the same sharing scheme to SVFT** so that both methods use a comparable trainable-parameter budget. The results in the table below show that PiCa’s projection contributes to improved performance compared to SVFT.
>
> | Method             | #Params | GSM-8K | MATH  |
> |--------------------|---------|--------|-------|
> | PiCa (r=256)       | 5.37M   | 52.77  | 16.36 |
> | SVFT_R w/ sharing (d=280)     | 5.48M   | 50.49  | 15.86 |
>
> &nbsp;
>
>
> **Q1**: From an algorithmic perspective, Theorem 2 extends this result to Algorithm 1 by showing that sequential gradient projections can approximate the projection of the accumulated gradients in Theorem 1. The Lipschitz assumption in Theorem 2 follows a standard analytical technique widely used in deep-learning theory.
>
>
> &nbsp;
>
>
> **Q2**: Thanks for raising the important point. We referenced the results of Full FT from [1]. Based on your insights, we ran the Full FT experiments ourselves and found that, with sufficiently low learning rates (1e-6/5e-6), Full FT achieves higher accuracy than PEFT methods as shown in the table below. We suspect SVFT used higher learning rates (1e-5/5e-5) for Full FT.
>
> [1] Lingam, Vijay Chandra, et al. "SVFT: Parameter-efficient fine-tuning with singular vectors." Advances in Neural Information Processing Systems 37 (2024): 41425-41446.
>
> | Setting                               | Gemma-7B (GSM8K) | Gemma-7B (MATH) | LLaMA-3-8B (GSM8K) | LLaMA-3-8B (MATH) |
> |---------------------------------------|------------------|------------------|---------------------|--------------------|
> | Reported in SVFT Paper                | 74.67            | 25.70            | 64.13               | 16.24              |
> | Our Reproduction (Low LR)   | 78.09            | 30.98            | 76.57               | 26.12              |

---

### Official Review · Reviewer_e2mp · 2025-11-04

**Soundness:** 2
**Presentation:** 3
**Contribution:** 2
**Rating:** 4
**Confidence:** 4

**Summary:**

This paper introduces PiCa, a new Parameter-Efficient Fine-Tuning (PEFT) method. The core idea is to improve parameter efficiency by providing a theoretically-grounded inductive bias. The method operates by projecting gradients onto the principal column space of the pre-trained weights, which the authors argue is the most effective subspace for adaptation. This gradient projection is then combined with a novel weight-sharing strategy, where layers with the same function (e.g., all "query" matrices) share a single set of trainable parameters, further reducing the parameter budget.


The authors provide a theoretical analysis to support this approach. Theorem 1 aims to justify that the principal column space captures the ideal update, based on the observation that fine-tuned weights are a small perturbation of the pre-trained weights . Theorem 2 argues that PiCa's practical "sequential projection" algorithm is a high-fidelity approximation of an "ideal" but computationally infeasible "accumulated projection" . Empirically, the paper demonstrates that PiCa achieves strong results on a wide array of NLP and vision tasks, often outperforming baselines like LoRA and SVFT with a significantly smaller parameter count

**Strengths:**

1. The paper tackles the problem of parameter-efficient fine-tuning, which is of critical importance for the practical adaptation and deployment of large-scale foundation models.

2. The paper is well-written and clearly structured. The authors effectively articulate the components of their method (gradient projection and weight-sharing) and the high-level intuition behind their theoretical claims.

3. The experimental evaluation is commendable for its broad coverage of foundation models. The method is tested across a diverse set of backbones, including both vision and language foundation models.

**Weaknesses:**

My main concerns lie with the theoretical justification, which feels less convincing than presented, and the comprehensiveness of the experimental baselines, which are insufficient to fully support the claims of state-of-the-art parameter efficiency.

1. The paper claims to provide a strong theoretical foundation, contrasting with prior "heuristic" SVD-based works. However, I am not fully convinced by Theorem 1's argument for projecting onto the column space. The justification relies on the finding that Full Fine-Tuning (FFT) produces a small deviation (gap) from the pre-trained weights . This "small deviation" is often an artifact of the FFT setup itself (e.g., low learning rates, few epochs) which is intentionally designed to prevent catastrophic forgetting. Thus, the argument feels somewhat circular: it observes that a "forgetting-constrained" method produces a small change, and then builds a theory based on this small-change assumption, rather than proving that the optimal unconstrained solution must lie in this space.

2. Both theorems provided upper bounds for the errors. These bounds look incomplete to me, in the sense that they are not proven to be optimal or tighter than bounds that could be derived for other projection (or even pruning) strategies. They do not, by themselves, demonstrate that this specific projection policy is inherently superior to any other.

3. The entire theoretical analysis is based on matrix norms (e.g., Frobenius norm). A small approximation error in the matrix norm does not necessarily or directly translate to higher parameter efficiency or better final task performance. This represents a significant gap between the theoretical claims and the practical outcomes.

4. The paper's core hypothesis—that the principal column space (the "head" of the weights) is the correct subspace for adaptation—is debatable. A growing body of work (e.g., [1], [2]) has suggested the opposite: that the tail singular vectors or the null space of the pre-trained weights may be more important for learning new task-specific knowledge while mitigating forgetting. The paper fails to acknowledge or address this contradictory line of research.


In addition, I feel that the experiments can be enhanced: While the backbone model diversity is a strength, the diversity of the PEFT baselines is a significant weakness. The paper claims state-of-the-art parameter efficiency but omits comparisons to many crucial, high-performance PEFT methods. To truly situate PiCa's efficiency, comparisons against methods like MiLoRA ([1]which focuses on the tail-space), PiSSA ([3] which also uses principal singular vectors), AdaLoRA ([4] which dynamically allocates rank), , and RoseLoRA ([5] which prunes LoRA entries in an unstructured way) are necessary. Without these baselines, the claim to superior efficiency is not fully substantiated.



[1] MiLoRA: Harnessing Minor Singular Components for Parameter-Efficient LLM Finetuning.

[2] AlphaEdit: Null-Space Constrained Knowledge Editing for Language Models.

[3] PiSSA: Principal Singular Values and Singular Vectors Adaptation of Large Language Models.

[4] AdaLoRA: Adaptive Budget Allocation for Parameter-Efficient Fine-Tuning.

[5] RoseLoRA: Row and Column-wise Sparse Low-rank Adaptation of Pre-trained Language Model for Knowledge Editing and Fine-tuning

**Questions:**

Please see my comments above.

---

> ### Author Response · Authors · 2025-11-23
>
> We thank the reviewer for their valuable feedback. We address the weaknesses (**W**) raised by the reviewer below.
>
> &nbsp;
>
> **W1**: **It is not that a “forgetting-constrained” method produces a small change. Rather, the optimum itself tends to lie in a small-update.** While the reviewer's concern about circularity comes from viewing the small-deviation as something artificially imposed to prevent forgetting, in practice it emerges as a prerequisite for reaching good optima when adapting large foundation models that are pre-trained on vast, diverse corpora. As shown in the table below, when sweeping learning rates in full fine-tuning, runs that move far from $W_0$ under large learning rates yield suboptimal performance. The best validation solutions consistently occur at sufficiently small learning rates and remain close to $W_0$, as illustrated in Fig. 2 ad Table 10 of our paper. In this context, Theorem 1 formalizes how projecting onto the dominant column space yields a near-optimal low-rank approximation.
>
> | Model      | LR        | GSM8K | MATH  |
> |--------------|-----------|-------|-------|
> | Llama3-8B      | 5.0e-05   | 64.90 | 17.80 |
> |              | 1.0e-05   | 74.68 | 24.56 |
> |              | 5.0e-06   | **76.57** | **26.12** |
>
>
> &nbsp;
>
> **W2, W3**: **Our theorems are not intended to prove the optimality or universal superiority of our projection over all other strategies, nor to claim that projection alone guarantees task-optimal performance.** Rather, our theorem shows that, in general, the dominant directions of the resulting $\Delta W$ can be well approximated within the pretrained column space $U_r$ of $W_0$. This means that by keeping $U_r$ fixed and learning only a small set of parameters  ($ \theta_f $) that determine the task-specific choice of how to move inside this space, we can reduce the number of trainable parameters, which is the notion of parameter efficiency we target. Different projection spaces may also reach good optima, which does not contradict our claim. Empirically, our experimental results show that leveraging this space allows us to use fewer parameters while achieving similar or better performance than different PEFT methods based on random initialization or other heuristic projections, suggesting that this is a useful and effective choice in practice.
>
> &nbsp;
>
> **W4**: **The previous works (MiLoRA and AlphaEdit) do not contradict our column-space.** MiLoRA initializes LoRA with tail singular vectors but does not freeze them. As a result, the learned $\Delta W$ is free to drift into essentially any subspace. AlphaEdit is even further from being contradictory. AlphaEdit aims to address knowledge editing, and its null space is defined with respect to preserved-knowledge keys ($K_0K_0^{\top}$) rather than the null space of the model’s pre-trained weight matrix $W_0$. Thus, AlphaEdit’s null space is not orthogonal to the principal column space of PiCa.
>
> &nbsp;
>
> **W5**: **Following the reviewer’s suggestion, we additionally evaluated MiLoRA, PiSSA, AdaLoRA, and RoseLoRA** on the Gemma-2B as shown in the table below. These additional results substantiate PiCa’s parameter efficiency compared to the other baselines.
>
> | Method              | #Params | GSM-8K | MATH  |
> |---------------------|---------|--------|-------|
> | Full-FT             | 2.5B    | 52.69  | 17.94 |
> | LoRA (r=32)         | 26.2M   | 43.06  | 15.50 |
> | MiLoRA (r=32)       | 26.2M   | 49.81  | 15.64 |
> | PiSSA (r=32)        | 26.2M   | 51.63  | 14.78 |
> | AdaLoRA (r=32)      | 26.2M   | 43.37  | 16.26 |
> | DoRA (r=16)         | 13.5M   | 44.27  | 16.18 |
> | RoseLoRA (r=32)      | 6.57M   | 44.67  | 16.24 |
> | SVFT_R (d=16)       | 6.35M   | 50.03  | 15.56 |
> | PiCa (r=256)        | **5.37M**   | **52.77**  | **16.36** |

---

### Meta-Review · Area_Chair_6Rdh · 2026-01-06

**Summary:**

The paper introduces PiCA, a new PEFT method where the key idea is to project gradient updates onto the top-rank singular vectors.  PiCa also shares the projected update matrix across layers of the same type which allows to reduce the number of parameters. In a parameter-matched setting, this allows PiCA to then increase the projection rank compared to other PEFT methods which is where the performance gains likely come from in my view. On suggestion of the reviewers, the authors compare to SVFT with weight-sharing, showing that PiCa still outperforms it. However, I think a fair comparison would apply the same weight-sharing scheme to other PEFT baselines as well, and I encourage the authors to do so. Moreover, it only compares to one SVFT variant in this setting and it's not clear that this is the best choice.

The theoretical analysis crucially hinges on the assumption that fine-tuning does not drastically change the singular vectors. While the authors provide some evidence that this holds in practice, especially for full fine-tuning with a low learning rate, this assumption should be made more explicit in the final version. In any case, while the theory is not particularly sophisticated or technically deep, it does provide some support and justification for why this particular projection can work well. This is sufficient for a paper of this nature in my opinion.

While all four reviewers gave the paper a rating of 4 "marginally below the acceptance threshold. But would not mind if paper is accepted" I believe that the authors sufficiently addressed the raised concerns and that at least some of the reviewers would have increased their rating (see below). Therefore, given that above, and given that the experimental evaluation is comprehensive, I recommend acceptance of the paper.

**Reviewer Concerns:**

The authors sufficently addressed almost all questions and issues raised by the reviewers, including among others:
- Reviewer UoTk: Clarification on the difference between PiCa and SVFT
- Reviewers e2mp, 8LsH: The assumptions behind Theorem 1 and the core claims
- Reviewer e2mp: Additional comparisons to more baselines
- Reviewer e2mp: Correct subspace for adaptation

Reviewer UoTk and 8LsH both raised the concerns that the performance stems mostly from the weight sharing, and requested that this strategy is also applied to the baselines. The authors did so but only for SVFT (and only for one variant). Applying weight sharing should be added to the other baselines as well in the final version of the paper.

The authors did not include additional results on OOD (as requested by Reviewer 8LsH) and long-context (as requested by Reviewer XBE4) task. While, I agree with the reviewers that this would indeed be helpful and strengthen the paper, I do no think it is necessary. Especially, since some of these additional experiments would require much more care and/or computational resources.

**Reviewer Scores:**

I believe Reviewer UoTk, 8LsH, and e2mp would have increased the score.

---

### Decision · Program_Chairs · 2026-01-26

Accept (Poster)